# Aldosterone: Essential for Life but Damaging to the Vascular Endothelium

**DOI:** 10.3390/biom13061004

**Published:** 2023-06-17

**Authors:** Michael Crompton, Laura J. Skinner, Simon C. Satchell, Matthew J. Butler

**Affiliations:** Bristol Renal, Dorothy Hodgkin Building, University of Bristol, Whitson Street, Bristol BS1 3NY, UK

**Keywords:** aldosterone, mineralocorticoid receptor, endothelium, glycocalyx

## Abstract

The renin angiotensin aldosterone system is a key regulator of blood pressure. Aldosterone is the final effector of this pathway, acting predominantly via mineralocorticoid receptors. Aldosterone facilitates the conservation of sodium and, with it, water and acts as a powerful stimulus for potassium excretion. However, evidence for the pathological impact of excess mineralocorticoid receptor stimulation is increasing. Here, we discussed how in the heart, hyperaldosteronism is associated with fibrosis, cardiac dysfunction, and maladaptive hypertrophy. In the kidney, aldosterone was shown to cause proteinuria and fibrosis and may contribute to the progression of kidney disease. More recently, studies suggested that aldosterone excess damaged endothelial cells. Here, we reviewed how damage to the endothelial glycocalyx may contribute to this process. The endothelial glycocalyx is a heterogenous, negatively charged layer on the luminal surface of cells. Aldosterone exposure alters this layer. The resulting structural changes reduced endothelial reactivity in response to protective shear stress, altered permeability, and increased immune cell trafficking. Finally, we reviewed current therapeutic strategies for limiting endothelial damage and suggested that preventing glycocalyx remodelling in response to aldosterone exposure may provide a novel strategy, free from the serious adverse effect of hyperkalaemia seen in response to mineralocorticoid blockade.

## 1. Introduction

The development of sodium and potassium gradients between the intra and extra cellular compartments was a key evolutionary step that facilitated life on earth [1]. Without these key electrochemical gradients, events within the cell cycle, such as transcription and translation, would not be possible. To allow terrestrial life, complex systems evolved to conserve salt and water and maintain homeostasis within the intra and extra cellular compartments [1]. This regulation is achieved, in part, by the renin angiotensin aldosterone system (RAAS). All the major components of this system were highly conserved through evolution and were present in ancestral chordates [1]. Aldosterone production is the final step in this regulatory pathway and the most potent stimulator of sodium conservation in mammals. Here, we review how aldosterone is produced and the regulatory steps that control its release and downstream signalling pathways. In addition to the biological role of aldosterone, we discuss identified pathological pathways implicated in human disease, highlighting that endothelial damage may be key. Finally, we highlight how damage to the endothelial glycocalyx may explain key pathological findings before discussing current therapeutic strategies and areas for future research.

## 2. Aldosterone Production

The enzyme renin is released in response to reduced renal (glomerular) perfusion pressure and in response to reduced sodium concentrations within the tubular lumen or sympathetic nervous system activation. Renin hydrolyses angiotensinogen to angiotensin I which is subsequently activated by angiotensin converting enzyme (ACE) to form angiotensin II. Aldosterone is produced in response to angiotensin II or elevations in extracellular potassium levels. Under physiological conditions, aldosterone synthesis has a circadian rhythm that parallels cortisol in humans [2] and corticosterone in rodents [3].

Aldosterone is produced in the zona glomerulosa of the adrenal cortex. Adenomas in the adrenal cortex or bilateral adrenal hypertrophy can both cause primary hyperaldosteronism resulting in hypertension and hypokalaemia. However, beyond the hypertensive effect, primary hyperaldosteronism increases the risk of cardiovascular events and proteinuria, even when compared to appropriate hypertensive controls [4,5,6,7]. Secondary hyperaldosteronism occurs in a number of common conditions, including idiopathic hypertension [8], obesity [9,10], advanced renal failure [11], obstructive sleep apnoea [8], and sleep disorders including shift work [8]. These patient groups also have an increased risk of developing proteinuria and cardiovascular disease [12,13].

Fluctuations in circulating aldosterone levels are largely the result of variations in de novo synthesis. Steroidogenesis starts with the “early regulatory step” of translocation of cholesterol to the inner mitochondrial membrane [14]. Translocation is facilitated by steroidogenic acute regulatory protein (St AR Protein) and alterations in this step can happen within minutes [14]. The “late regulatory step” in aldosterone production is mediated largely through modulation of expression of the enzyme aldosterone synthase (CYP11β-hydroxylase) (CYP11B2 human) [14]. The direct regulation of this enzyme by circulating factors outside of the RAAS may help explain variations in serum aldosterone levels. Leptin, a hormone produced by adipose tissue, regulates hunger and energy balance, and increases circulating aldosterone levels. Leptin acts, via its own receptor on CYP11β-hydroxylase, modulating a calcium-dependent regulatory mechanism to increase aldosterone production [15]. This finding may help to explain why certain clinical groups have high circulating aldosterone levels even in the absence of ‘traditional’ RAAS stimuli and a diminished response to angiotensin blockade. The discovery of leptin as a direct stimulus for aldosterone production also opens the possibility that there may be other circulating factors capable of stimulating aldosterone release.

The role of extra adrenal aldosterone production remains debated. There is some evidence that cardiac myocytes [16,17], pulmonary vascular endothelial cells (EnC) [18], systemic vascular EnC [19], and cells within the CNS [17] may be able to produce aldosterone in quantities capable of eliciting a local effect. However, recent work has failed to demonstrate significant aldosterone synthesis within the vascular endothelium, and so, its effect within these specialized cells is thought to be limited [20]. Adipocytes surrounding blood vessels may produce a basal level of aldosterone and respond to stimulation with angiotensin II with further increased production [21]. Blocking the effect of locally produced aldosterone within mesenteric arteries from diabetic *db*/*db* mice was shown to increase the degree of the vasodilation seen in response to acetylcholine. The authors concluded that in the context of diabetes, adipocyte derived aldosterone may act in an autocrine and paracrine manner to regulate adipocyte differentiation and vascular function, respectively [21]. Similarly, aldosterone may act in a paracrine manner to potentiate localised inflammation. In Wistar rats, local aldosterone production by aldosterone synthase in peripheral sensory neurons activated neuronal mineralocorticoid receptors (MR), contributing to inflammation-induced mechanical hypersensitivity; this effect was attenuated with selective MR blockade [22]. A consensus view on the importance of extra adrenal aldosterone production clinically is yet to be reached and remains an area of ongoing research.

## 3. Aldosterone Acts on Multiple Tissues and May Act via Multiple Receptors

The classical role of aldosterone is to conserve sodium. This action is achieved within the distal tubules and collecting ducts of the kidney via stimulation of epithelial cytosolic MR. Binding of aldosterone to MR dislodges chaperone proteins from the receptor to form an aldosterone-MR complex. This complex trans-locates, from the cytosol to the nucleus, where it alters target gene transcription. Increased serum and glucocorticoid-induced kinase 1 (SGK-1) leads to altered expression of amiloride-sensitive epithelial sodium channels (ENaC). These channels migrate to the cell membrane and facilitate the conservation of sodium.

The above pathway was well studied, but in recent years, it became apparent that MR expression in other tissues is common. MR are widely expressed in epithelial tissues including the bowel and salivary glands [23], but also in non-epithelial tissues such as vascular smooth muscle [24,25,26,27], skeletal muscle [28], cells of the immune system [29], vascular EnCs [24,30,31,32,33,34], cardiac myocytes [35,36], and adipocytes [37]. Our group confirmed that even within the human glomerulus, EnC and podocytes express MR [38]. The function of these receptors, in cells distant from the renal tubules, is not yet fully understood.

Aldosterone and the much more prevalent glucocorticoids, such as cortisol, have an equal affinity for the MR binding zone. Specificity is achieved in a tissue-specific manner though the expression of 11 –beta-hydroxysteroid dehydrogenase type 2 (11 β-HSD2) [39]. This enzyme, even when present at very low levels, metabolizes glucocorticoids to their metabolically inactive substrates [40]. 11 β-HSD1 in contrast can catalyse this reaction in either direction but generally favours the generation of active cortisol from inactive cortisone. Thus, the balance in activity of these two enzymes within the cytosol dictates the level of MR stimulation by circulating glucocorticoids. In mice, deficiency of 11 β-HSD2 in the vascular endothelium increased inflammatory markers within the endothelium and accelerated atherogenesis in ApoE-/-mice [41]. The MR antagonist eplerenone reversed this effect [41]. Knock out of 11 β-HSD2 in rats renders animals polydipsic, polyuric, and proteinuric with histological changes within the kidneys that are consistent with chronic kidney disease [42]. In human umbilical vein EnC (HUVEC), cortisol was noted to reduce EnC nitric oxide synthase (eNOS) production [43]. This effect was markedly increased following pre-treatment of cells with small interfering RNA (siRNA) targeting 11 β-HSD2, an effect that could be attenuated with spironolactone. These results suggest that knock down of 11 β-HSD2 allowed cortisol to remain active in the cytosol, modulating eNOS via MR [43]. Down regulation of 11 β-HSD2 was also linked with salt sensitive hypertension in human linkage studies and animal models [39,44,45]. Unilateral nephrectomy, in animal models, reduced 11 β-HSD2 expression within the remaining kidney and predisposed animals to a salt responsive hypertension [39,46,47]. In humans, a number of studies demonstrated a prolonged plasma half-life for cortisol in patients with CKD. A study of 95 adult renal patients also demonstrated a direct correlation between the expression of 11 β-HSD2 and creatinine clearance [48]. In humans, we know that liquorice ingestion (which inhibits 11 β-HSD2) or functional polymorphisms in 11 β-HSD2 expression can predispose individuals to a salt sensitive form of hypertension, but the full effect of reduced functioning renal mass on this complex system is yet to be studied [49,50].

Aldosterone does not act solely via modulation of nuclear transcription of target genes (Figure 1). ‘Nonclassical’ receptors including GPER were also demonstrated to be aldosterone-responsive [51]. The first detectable response to aldosterone occurs within minutes of exposure. This non-classical response result in aldosterone activating enzymes within the cytosol and extra nuclear compartments within minutes to activate protein kinases [52]. Activated kinases subsequently phosphorylate membrane channels and secondary enzymatic targets to modulate sodium and potassium channel activity or availability and pro-inflammatory pathways [51,53]. Aldosterone also stimulates receptors located at the vascular EnC surface [54]. The isolation of MR in the cell membrane fractions from Wistar rats and human embryonic kidney cells (HEK 293) adds further weight to the argument that cell surface MR exists and may be functional [55,56]. In rats, the function of surface receptors was studied using pegylated aldosterone. Pegylated aldosterone is incapable of crossing cell membranes but still activates ERK 1/2 phosphorylation by binding to receptors on H9c2 cells surface. In Sprague Dawley rats, activation of these surface receptors did not increase infarct size (in contrast to non-pegylated aldosterone). Spironolactone did not prevent ERK1/2 phosphorylation but the G-protein associated oestrogen receptor (GPER or GPR 30) inhibitor G36 did prevent phosphorylation, suggesting that aldosterone may activate GPER on the cell surface [57]. Despite being labelled as an oestrogen receptor, GPER is sensitive to aldosterone at very low doses [51]. This field remains controversial; however, due to a lack of evidence of competitive aldosterone binding; thus, not all conditions to define GPER as an aldosterone receptor were met [58]. GPER on a breast cancer-derived EnC line, for example, suggested that aldosterone does not bind to GPER; however, it does induce a direct interaction between MR and GPER and between GPER and epidermal growth factor receptor (EGFR). In this study, the presence and activation of both MR and GPER was required for proliferation and migration of breast cancer cells to occur [59]. Whilst these findings need to be validated in other cell lines, they open the possibility that the net effect of aldosterone exposure on different tissues may be dependent on the variable expression of both MR and GPER. However, although the status of GPER as an aldosterone receptor remains uncertain, several groups found that GPER seems to be involved in aldosterone signalling. A recent work proposed a novel autocrine-paracrine GPER-mediated mechanism of aldosterone synthesis in human adrenocortical cell lines (HAC15). Incubation of HAC15 with aldosterone increased expression of aldosterone synthase (CYP11B2), an effect unchanged by addition of the MR antagonist canrenone. In contrast, antagonism of GPER by G36 abolished this increase in CYP11B2 expression, suggesting that aldosterone acts via GPER to enhance CYP11B2 gene expression [60]. Rat aorta EnC were used to study the effects of GPER stimulation due to their unique lack of MR but consistent GPER expression [61]. In this preparation, GPER stimulation mediated pro-apoptotic and anti-proliferative effects as well as vasodilatation [61]. The vasodilator effect seen was in contrast to the action of aldosterone via MR, where, generally, vasoconstriction is considered the predominant effect resulting from impaired NO production [62]. Similarly, seemingly opposing effects of GPER versus MR activation were recently reported in rat cardiomyocytes. Aldosterone acting via MR-induced hypertrophy of myocytes. GPER activation in the presence of G1 (a specific GPER agonist) and aldosterone halted this MR-mediated hypertrophy [63].

The complex interaction between 11 β-HSD1/2 and recent discoveries of aldosterone’s rapid actions, cell surface receptor, and potential action via GPER all add additional layers of complexity to the process of aldosterone signalling in the vascular endothelium. They also present exciting new targets through which the pathological effects of aldosterone on the vascular endothelium may be modulated.

## 4. The Vascular Endothelium

EnC form the innermost cell layer of the vascular system. These specialist cells adapted to function optimally within different areas of the vascular tree where they contribute to the regulation of fluid filtration, vessel tone, coagulation, neutrophil recruitment, and hormone trafficking [64].

In the last few decades, researchers showed that the EnC membrane is not the innermost structure of blood vessels. Lying on the luminal aspect of the EnC is an anionic biopolymer layer called the glycocalyx. This complex structure consists of components anchored to the cell surface such as proteoglycans and sialoproteins, along with elements adsorbed from the circulation including albumin [65]. The anionic charge is largely due to the expression of the glycosaminoglycans heparan sulphate (HS) and chondroitin sulphate(CS). The integrity of the overall structure seems to be dependent on both fixed and adsorbed components. Many of the specialist functions performed by the endothelium are dependent on a ‘healthy’ glycocalyx [65]. There is growing evidence that this innermost layer can be affected negatively by MR stimulation [66].

## 5. Aldosterone Disrupts Flow-Mediated Dilatation 

Flow-mediated dilatation (FMD) is defined as the EnC-dependent process facilitating the relaxation of a vessel in response to shear stress. FMD occurs as EnC respond to the movement of fluid over their luminal surface by producing NO. EnC dysfunction, detected by altered FMD, is an important independent risk factor for cardiovascular disease and remains clinically significant once adjustments are made for underlying common pathologies ,including diabetes [67,68], smoking [69,70], and hypertension [71].

Patients with high circulating levels of aldosterone have impaired EnC-dependent FMD [72]. In addition, elevated levels of biomarkers of EnC dysfunction (von Willebrand Factor, ICAM-1, ox-LDL) were seen in hyperaldosteronism [6]. Patients with a relative excess of aldosterone, compared to renin levels, also have impaired NO-mediated vasodilatation [73]. Recent work found FMD was significant reduced in patients with primary hyperaldosteronism as compared to those with essential resistant hypertension [74]. Fortunately, some reports suggest EnC dysfunction due to MR stimulation is reversible. Spironolactone increases EnC-dependent flow-mediated dilatation in patients with heart failure, even when used in addition to other medical treatments [75]. However, in a recent randomized controlled trial (RCT), spironolactone failed to improve FMD in early autosomal dominant poly cystic kidney disease (ADPKD) patients. Although the small sample size (n = 60) and the probable absence of significant EnC dysfunction in this cohort (those with significant co-morbidities and/or renal impairment, eGFR < 60, were excluded) may explain the results observed [76]. 

The eGlx is integral to the process of FMD [77,78]. To date, at least ten candidate mechanosensitive molecules were identified within the eGlx [78]. Conflicting data emerged from enzymatic degradation studies and gene deletion studies, making isolation of the most important mechanosensitive molecules difficult [79]. HS and syndecan-4 remain important candidate molecules. Our recent work in a type 1 diabetic mouse model suggests that eGlx damage occurs through matrix metalloproteinase (MMP)-induced syndecan-4 shedding. Treatment of mice with an MMP inhibitor restored eGlx, reduced albuminuria and glomerular permeability, thus presenting MMP inhibition as an attractive, novel therapy in those with diabetic kidney disease [80]. 

## 6. Aldosterone Increases Permeability to Macromolecules

The vascular endothelium plays a key role in regulating the movement of macromolecules from the blood into the tissues. Aldosterone, at a dose of 10^−9^ mol/L, increased the EnC permeability to proteins <70 Kda within 60 min both in cultured HUVECs and ex vivo arteries [81]. In contrast, the permeability of HUVECs was found to be unaffected following 3 days exposure to aldosterone [82]. It seems likely that the duration of aldosterone exposure is important. Whilst the study of systemic EnC permeability is complex, the impact of aldosterone and MR inhibition on the renal glomerulus was studied by several groups, including our own. Protein largely enters the urine in significant quantities following increased glomerular protein permeability [65,83]. The glomerular ‘sieve’ is composed of multiple layers, all of which are vital to the integrity of the glomerular filtration barrier (Figure 2). 

The EnC were once thought to have a minimal impact on the pathogenesis of proteinuria. However, more recent research highlighted the importance of the EnC, and the glycocalyx they produce, in regulating glomerular protein passage [65,85]. Both primary and secondary hyperaldosteronism in humans are linked to proteinuria [6,12,13,86,87]. An MR blockade was shown to be effective in reducing urinary protein leak [87]. Reducing urinary protein loss through the glomerular filtration barrier reduces patients’ risk of progressive kidney failure and is a key strategy for managing chronic kidney disease [88]. The reduction in proteinuria in response to MR blockade is independent of blood pressure reduction and occurs even when MR blockade is used in addition to other RAAS blocking agents [38,87,89,90,91]. Further evidence for this blood pressure independent effect was gathered by normalizing blood pressure (with agents not targeting RAAS components), which failed to prevent renal fibrosis and proved ineffective at reducing urinary protein loss [92,93]. We showed that spironolactone, even when used in addition to ACE inhibition, reduced the levels of circulating glycocalyx-degrading enzymes in patients with diabetes. In addition, our study highlighted that MR inhibition in a diabetic rat model preserved the glycocalyx and reduced albuminuria, again by reducing glycocalyx degrading enzyme activity and preserving the EGlx [84]. Podocytes also express MR and are damaged by elevated aldosterone [94]. However, podocyte-specific MR knock out mice still become proteinuric and systemic aldosterone blockade in these animals remains effective in reducing glomerular injury, suggesting other cell types are important in the pathogenesis of proteinuria in response to MR stimulation [95].

It is possible that crosstalk between cells of the renal glomerulus in response to aldosterone stimulation contributes to the development of proteinuria. Aldosterone acts via MR to increase endothelin-1 (ET-1) expression in Sprague Dawley rat kidney tissue [96] Endothelin-1 increases heparanase release from podocytes and this enzyme is known to cleave heparan sulphate, a major component of the glycocalyx, loss of which is associated with proteinuria [65,97]. Conversely, Atrasentan, an endothelin A receptor antagonist, increased eGlx coverage and reduced albuminuria in diabetic mice [98]. The RCT (SONAR), showed that atrasentan reduced CKD progression in type 2 diabetics; however, there was a greater incidence of heart failure in the treated group (despite careful selection in a pre-randomisation enrichment period designed to minimise this risk). Whilst this effect was not statistically significant between groups, the trend suggests that widespread use of endothelin receptor antagonists may be limited by the risk of heart failure associated with these medications [99]. 

The increased permeability of the systemic endothelium in response to aldosterone, and the associated glycocalyx remodelling, hint towards the importance of the glycocalyx in the pathogenesis of proteinuria in response to aldosterone excess. The possibility of glomerular eGlx protection as a potential pharmacological target in proteinuric renal disease will rely on ongoing research to elucidate the steps in this complex process.

## 7. Aldosterone Promotes Inflammation

The ability of white blood cells to egress from the vascular compartment to a site of tissue injury or infection is an essential part of the innate immune system (Figure 3). 

However, when the regulation of this process becomes defective, unwanted inflammation and fibrosis may occur. Aldosterone exposure rapidly increases the adhesion of polymorphonuclear leukocytes to the HUVEC surface [100]. These effects were maximal within one hour of aldosterone exposure. Increased protein expression of vascular cell adhesion molecule (VCAM)-1, e-selectin, and intercellular adhesion molecule-1 (ICAM-1) were thought to be responsible for the effect [100]. In ApoE knockout mice, who received aldosterone and an atherogenic diet, ICAM-1 expression increased, as did atherosclerotic plaque size; this was not seen in ApoE/I-CAM double knockouts [101]. MR deletion from endothelial cells prevented adverse cardiac remodelling in mice in response to the MR agonist deoxycorticosterone acetate (DOCA) in the presence of high salt [102]. This was associated with a reduced expression of the VCAM-1 and ICAM-1 in MR-deficient endothelial cells and less macrophage infiltration of the heart [33]. In human coronary EnC, ICAM-1-dependent leucocyte adhesion was inhibited in MR-knockdown cells and by spironolactone [103]. Taken together, these implicate aldosterone-induced ICAM-1 expression in the progression of atherosclerosis, an effect that appears inhibited by MR blockade.

It is important to note that the expression of cell adhesion receptors is also regulated by shear stress [104]. As highlighted previously, the glycocalyx is intimately involved in this process and so, damage to the glycocalyx structure, which limits the EnC protective response to shear stress, may also result in the cell surface becoming more adhesive [104] (Figure 3). These data, when combined with the observation by Oberleithner (that the glycocalyx becomes thin and loses heparan sulphate from its structure in response to aldosterone and salt exposure) [66,105], suggest that the transit of white blood cells from the circulation may increase in response to aldosterone. This may be due to increased adhesion molecule expression, loss of shear sensitivity, and increased adhesion molecule exposure due to reduced expression of other glycocalyx components such as HS. 

## 8. Aldosterone Promotes Atherosclerosis

Elevated aldosterone levels, even within the ‘normal range’, are strong predictors of cardiovascular disease risk [106]. This effect remains even when adjusted for baseline atheroma [107]. Atherosclerosis is a systemic vascular inflammatory disease initiated by cardiovascular risk factors that cause EnC damage [108]. Activated EnC recruit leukocytes to the EnC wall, typically in areas of low shear stress [108]. High circulating levels of aldosterone are linked to an increased rate of atherogenesis, particularly in areas of the circulation that experience low shear stress, e.g., vessel bifurcations, etc. In a mouse model with elevated circulating aldosterone at levels insufficient to cause BP alteration, aldosterone increased recruitment of activated monocytes and T cells to atheroma prone areas, promoting atheroma formation [24]. A recent work in a mouse model of low shear stress (via partial carotid artery ligation) proposed that AMPK inactivation increases Na^+^ -H^+^ exchanger (NHE)1 activity and downstream hyaluronidase (HYAL2) 2-mediated glycocalyx degradation. Activation of AMPK with ampkinone reduced HYAL2 activity and halted glycocalyx loss; blocked ICAM-1 and VCAM-1 expression and reduced macrophage recruitment [109]. As discussed previously, aldosterone is implicated in ICAM-1 expression; however, the exact contribution to this signalling pathway and associated glycocalyx loss remains unknown.

## 9. Pharmacological Manipulation of the Aldosterone—MR System

Although we are only now beginning to understand how aldosterone excess results in tissue damage, the clinical use of MR blocking drugs was widespread since their introduction in 1953. The benefits of MR inhibition in humans, beyond BP control, was established following two landmark studies. The first study was performed by Pitt et al. and was designed to review the effects of spironolactone administration on the morbidity and mortality of patients with heart failure [110]. The second study, again by Pitt et al., looked specifically at patients with left ventricular dysfunction after a myocardial infarction [111]. Both studies reported significantly improved clinical outcomes in patients receiving spironolactone. Since then, the discovery of multiple pathological pathways influenced by MR and the clinical benefits associated with MR inhibition have driven further drug discoveries targeting the RAAS system.

### 9.1. Angiotensin Inhibition

Blockade of the RAAS at any level was initially thought to inhibit aldosterone secretion to acceptable levels. Both angiotensin-converting enzyme inhibitor (ACEi) and angiotensin receptor blocker (ARB) medication reduce serum aldosterone levels initially [112,113]. However, the phenomenon of aldosterone escape or breakthrough is now recognized clinically. The incidence of aldosterone escape depends on the definition used. The most widely adopted definition is “an increase in plasma aldosterone levels during long-term RAAS blockade, not compared to pre-treatment levels but to aldosterone levels after 2 months treatment” [113].

Using this definition, researchers in Denmark found 41% of type 1 diabetic patients on long term ARBs suffered from aldosterone escape [113]. Sato et al. described a similar incidence of aldosterone escape in hypertensive patients and type 2 diabetic patients [114,115]. Thus, even in the presence of maximal ACEi or ARB therapy, aldosterone may not be fully suppressed. Their use clinically is also limited by their recognized side effects. These include hyperkalaemia, an increased predisposition to acute kidney injury (AKI) [116], hypotension, teratogenicity, and cough.

### 9.2. Mineralocorticoid Antagonism

The MR antagonist Spironolactone was originally developed as a potassium sparing diuretic for the treatment of oedema. Its usefulness in primary hyperaldosteronism was subsequently discovered. More recently, as the benefits of MR blockade in heart disease became more widely recognized [87,110,111], the use of spironolactone in the UK greatly increased [117]. Spironolactone is readily absorbed with a bioavailability of 80–90% [118]. It has a half-life of only 1.4 h due to metabolism within the liver; however, the major metabolite (7 α-thiomethylspironolactone) is also pharmacologically active and accounts for 80% of the effects of the parent compound. Because of these effects, spironolactone can be administered once daily but it takes several days to reach steady state and several weeks to reach its maximal hypotensive effect [118]. Spironolactone has a relative binding efficacy for MR of 0.11 (Aldosterone = 1) but is not selective for MR blockade [118]. The non-specific binding of spironolactone accounts for the sexual side effects including gynaecomastia, mastodynia, impotence, and menstrual irregularities. These effects occur because of spironolactone’s effect on progesterone receptors (PR) and androgen receptors (AR). Spironolactone’s use is also limited by the predictable retention of potassium that results from aldosterone blockade. This becomes an issue in patients with impaired potassium regulation. For this reason, spironolactone must be used with extreme caution in patients with chronic kidney disease (CKD). The BARACK-D trial should give us valuable information regarding the safety of spironolactone in patients with CKD and its results are eagerly awaited now that recruitment has ended [119]. 

Eplerenone, a second-generation MR antagonist, became available for clinical use in 2002. Compared to spironolactone, eplerenone is a much more selective antagonist for MR [120]. However, the relative affinity of eplerenone for MR is lower (0.005) and with a plasma half-life of only 4–6 h (with no active metabolites) necessitating twice daily dosing. Eplerenone has a much more limited side effect profile due to its reduced affinity for progesterone receptors (140-fold lower) and androgen receptors (1200-fold lower) relative to spironolactone [120].

A third generation of novel non-steroidal MR antagonists (finerenone, esaxerenone and apararenone) were recently developed. Clinically, finerenone was now evaluated in two landmark clinical randomised control trials (FIDELIO-DKD and FIGARO-DKD) [121,122]. Experiments using radiolabelled drug suggest that finerenone may achieve a more even organ distribution concentration in contrast to spironolactone where renal accumulation was prominent [123]. The ARTS study reported a lower mean elevation in serum potassium in chronic kidney disease 3 (CKD3), chronic heart failure (CHF) patients on finerenone, as compared to those taking spironolactone [124]. The improved side effect profile was hypothetically linked to the shorter plasma half-life; however, hyperkalaemia remains a predictable effect of all MR antagonists, including finerenone and must be considered during clinical use [125]. 

### 9.3. Aldosterone Synthase Inhibitors

Blocking the MR receptor leads to further stimulation of the RAAS via potassium accumulation and blood pressure modulation. This, in turn, promotes higher levels of circulating aldosterone, which may eventually necessitate dose increases to maintain the treatment effect. At higher doses, cross reactivity of the MR antagonists with AR and PR became an increasing issue and the resulting side effects more common. Thus, drugs that inhibit aldosterone production/secretion are an attractive therapeutic option. Several agents were in development, including LCI 699 (osilodrostat). However, there are concerns about the non-selectivity of the 11-hydroxylation step [120]. Targeting this enzyme introduces the possibility of interference with the cortisol synthesis pathway, raising concerns about inducing an Addisonian crisis at times of high stress [120]. A phase III study involving 137 patients reported adrenal insufficiency in 28%, hypo-cortisolism in 51%, and adverse events related to adrenal hormone precursors in 42% of the participants [126]. The search continues for more specific molecules to block aldosterone synthesis with new models being established to facilitate this search [127,128]. Sakakibara et al. reported a synthesis of selective aldosterone synthase (CYP11B2) inhibitors with high potency and selectivity over 11β-hydroxylase (CYP11B1) that reduced aldosterone production in cynomolgus monkeys [129]. Sparks et al. reported the development of a selective aldosterone synthase (CYP11B2) inhibitor that reduced aldosterone production in cynomolgus monkeys without the accumulation of steroid precursors (11-doxycortisol, 11-deoxycorticosterone) seen with LCI 699 [130]. Most recently, a phase II trial of Baxdrostat demonstrated a promising human safety profile when used in patients with resistant hypertension [131]. This 12-week placebo-controlled trial confirmed a significant (additional 11 mmHg reduction in systolic BP) compared to control participants [131]. A longer trial including ambulatory BP is now needed to see how trial can be translated to long-term clinical practice.

## 10. Conclusions

The eGlx forms a vital, but fragile, component of the vascular system. Inappropriate levels of salt and aldosterone result in damage to the endothelium, leading to functional impairment of the eGlx. Pharmaceutical companies are investing large amounts of capital to find drugs capable of preventing the negative clinical impact of aldosterone excess. Newer agents will have improved side effect profiles and improved specificity for MR. However, the failure to look for new targets within this signalling cascade left us focused on MR, which has a physiological, highly conserved role in the regulation of total body sodium and potassium. Targeting MR will inevitably result in elevated serum potassium levels and may preclude the use of these agents in patients with kidney disease. Focusing on how the negative effects of aldosterone develop may reveal safer pharmacological targets in the battle to prevent EnC damage.

## Figures and Tables

**Figure 1 biomolecules-13-01004-f001:**
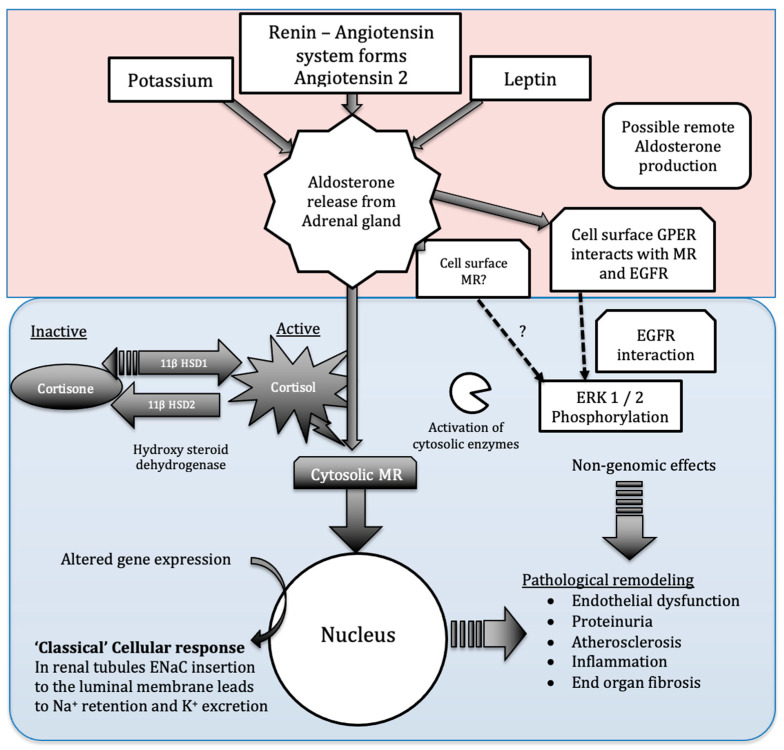
Multiple pathways coordinate and regulate aldosterone signalling. Multiple stimuli result in aldosterone production within the adrenal glands, the role of localized aldosterone production remains unknown. Circulating aldosterone my act on cell surface MR receptors and GPER in addition to the cytosolic MR receptor. The ‘classical’ aldosterone signalling pathway (via cytosolic MR) is protected from cortisol activation by hydroxysteroid dehydrogenase 2 activity. These complex pathways all represent potential future therapeutic targets.

**Figure 2 biomolecules-13-01004-f002:**
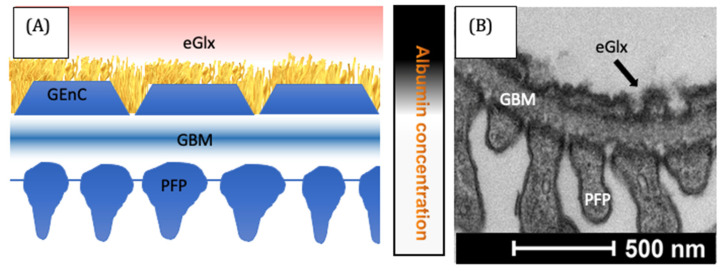
The glomerular filtration barrier. In the representative diagram, (**A**) the filtrate passes the layers of the filter. Albumin is largely excluded, as indicated by the local albumin concentration on the right. Electron microscopy (**B**) demonstrates the functional arrangement of cells within the glomerulus [84]. GBM = glomerular basement membrane, GEnC = glomerular endothelial cells, PFP = podocyte foot process.

**Figure 3 biomolecules-13-01004-f003:**
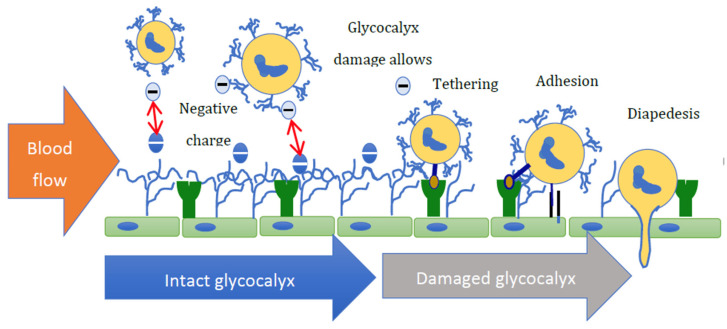
A suggested mechanism by which aldosterone may affect immune cell transit. In health, repulsion between the negatively charged glycocalyx of white blood cells and the vessel wall limits their interaction. Given the difference in height, the glycocalyx (∼500 nm) shields selectins (∼40 nm). Glycocalyx damage reduces the negative surface charge and allows for electrostatic attraction of white blood cells to the endothelium, which then mediate receptor–ligand interactions, resulting in tethering, rolling, adhesion, and diapedesis.

## Data Availability

Not applicable.

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
