# Peer review of "Aldosterone: Essential for Life but Damaging to the Vascular Endothelium"

_biomolecules, 2023, doi:10.3390/biom13061004_

Round 1

Reviewer 1 Report

The authors propose here a comprehensive review on the molecular actions of aldosterone with a particular focus on the hormone damaging effect to the vascular endothelium.

The text is well organized and documented, with relevant figures, and pleasant to read. I have only a few specific comments:

1) Line 57: CYP11beta-hydroxylase is only a part of the activity of the enzyme. To prevent any confusion the name of the gene (CYP11B2) or the term aldosterone synthase should be also mentioned here.

2) Fig. 2: The two boxes inside the cell are confusing because they do not correspond to cell compartments. They should be removed. The text on the left of the nucleus is truncated. It should be “ … Na+ retention and K+ excretion”.

3) Line 149 “aldosterone directly interacts with enzymes…”. I do not think it is true. To my knowledge, all aldosterone actions are mediated by a receptor, even the rapid non genomic actions restricted to the cytosol. Importantly, several non genomic responses to aldosterone are insensitive to blockade by spironolactone, but they are clearly prevented by eplerenone, a more selective MR antagonist. For example, the early (10 min) non genomic chronotropic action of aldosterone on isolated ventricular cardiomyocytes was completely abolished by eplerenone but not by spironolactone, while the later (24h) response was prevented by both antagonists (Rossier et al. 2012, Endocrinology 153:1269). Therefore, the absence of effect by spironolactone (as mentioned on line 151) does not necessarily means that MR is not involved.

Minor points:

- line 108: “manner” instead of manor

- initial and final page numbers are missing in several references : 1, 7, 16, 17, 25, 27, 29, 31, 38, 40, 47, 63, 65, 77, 78, 86, 95, 113, 116, 117, 119, 120.

Author Response

Many thanks for your review. We are glad you felt the review was comprehensive and well organised.

Addressing your points

  1. We have added the gene name and corrected the text as suggested
  2. Figure 2 has been updated to make the text easier to read and we have removed the boxes within the cell area as suggested, many thanks for highlighting the missing text.
  3. Many thanks for referring us back to look at the text in this area.  We agree with your comments and have tried to re-write this paragraph to make the findings clearer.
  4. Thank you for spotting manor vs manner - now corrected.  I have updated the endnote file attached to the manuscript, but I note the journal template is still not providing all page numbers. We will lease with the Journal staff about how to remedy.

Reviewer 2 Report

This is a comprehensive review summarizing the association between aldosterone and cardiorenal diseases, with particular focus on the emerging role of the endothelial glycocalyx. The manuscript is up-to-date and informative. This reviewer has only a few minor suggestions for the authors:

1. Page 10, “Aldosterone Synthase inhibitors” section: Please cite and briefly discuss the following article: Freeman et al. Phase 2 Trial of Baxdrostat for Treatment-Resistant Hypertension. N Engl J Med. 2023 Feb 2;388(5):395-405.

2. An abbreviation list would be helpful for the readers.

Author Response

We would like to thank the reviewer for their comments and we are glad they felt the review was already well written and comprehensive. We have updated the manuscript to highlight the recently published trial using Baxdrostat and we have added an abbreviation list for the editors to consider using.